# What We Don't C: Manifold Disentanglement for Structured Discovery

## Abstract

Accessing information in learned representations is critical for annotation, discovery, and data filtering in disciplines where high-dimensional datasets are common. We introduce *What We Don't C*, a novel approach based on latent flow matching that disentangles latent subspaces by explicitly removing information included in conditional guidance resulting in meaningful residual representations. This allows factors of variation which have not already been captured in conditioning to become more readily available. We show how guidance in the flow path necessarily represses the information from the guiding, conditioning variables. Our results highlight this approach as a simple yet powerful mechanism for analyzing, controlling, and repurposing latent representations, providing a pathway toward using generative models to explore *what we don't capture, consider, or catalog*.

## 1 Introduction

Representation learning aims to map a dataset to a lower-dimensional data manifold where each data point is represented by a lower-dimensional vector on this manifold. These representations have found many uses including data filtering (Liang et al., 2018), searching (Khattab & Zaharia, 2020), clustering (Ren et al., 2024), labeling (Asano et al., 2020), outlier detection (Han et al., 2022), visualization (McInnes et al., 2018; van der Maaten & Hinton, 2008) and more (Bengio et al., 2013). Approaches to representation learning are often validated on known features of the underlying data. Any methodological improvements are measured against benchmarks, such as how well a linear or non-linear model can recover certain features from the representations. In practice, approaches to representation learning often remain unchanged even when supervised labels are available.

In this work, we introduce *What We Don't C* (WWDC), an approach that disentangles known features from existing data manifolds. We make no requirement of our approach to fully separate all features into individual dimensions and instead aim only to separate features from a given manifold, enabling applications to real-world datasets and features. Conditioning on known features of a dataset, we use representations (e.g. from VAE; Kingma et al., 2014; Rezende et al., 2014) to flow match (Lipman et al., 2023) to a base distribution. Far from what is often just considered a base or 'noise' distribution, our work highlights that substantial structure is preserved from the representations. When flowing to the base distribution using conditioning guidance, features belonging to the conditioning are *suppressed*, which allows easier access to otherwise obfuscated features. Having identified its potential as a tool for scientific discovery, we illustrate how WWDC can be used in a discovery engine in Figure 1.

In this work we:

- Detail an approach which uses latent guided flow matching to enable structured discovery without conflating the most dominant signals that have already been thoroughly *captured*, *considered*, and *cataloged*, moving to uncover meaningful representations of *what we don't see*.

- Highlight theoretical arguments for the removal and preservation of relevant structures in the manifold.

- Validate our geometric understanding of conditional flows and their base distributions on a fully synthetic dataset.

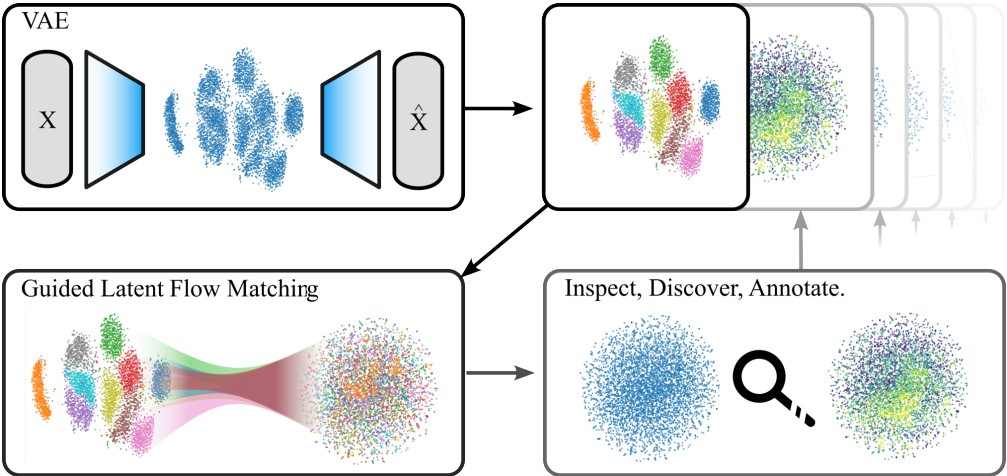

Figure 1: Scientific discovery of What We Don't C (WWDC). The representations of a VAE can be used to annotate data. These (now) known features can then be used to condition a flow, removing them from the manifold. This new disentangled manifold can then be inspected to uncover new features of the data. With sufficient coverage, these can then be used to continue the cycle, enabling access to further features.

- Verify the approach through controlled experiments on a colored variant of MNIST.
- Demonstrate the utility of this method for real-world datasets by directly isolating the disentangled features of real galaxy images.

Section 2 discusses and introduces the relevant background material. Section 3 explains the approach adopted in WWDC including why the resulting space maintains useful structures from the original manifold. Section 4 presents results across increasingly complex datasets and problems. These experiments span various degrees of complexity from simple 2D Gaussians in Section 4.1, a colored variant of MNIST in Section 4.2, and astrophysics galaxy morphology in Section 4.3. Section 5 summarizes the work and discusses future directions. Comprehensive details and additional generative samples are provided in the appendices.

## 2 BACKGROUND

### 2.1 MANIFOLD DISENTANGLEMENT

In this work, we adopt a different approach to disentanglement which we term manifold disentanglement. Using existing representations and known data features, we create new representations conditioned on the known features. This creates a rich, residual representation which we explore in Section 3 and Section 4. Notably, manifold disentanglement requires an existing manifold to disentangle, meaning this problem is defined for frozen, and pre-trained representation learning models. This cuts computational costs significantly, which is important for the envisioned application: the iterative discovery of new signals of interest in data as outlined in Figure 1.

This methodology differs from existing notions of disentanglement in the literature. In particular, unsupervised disentanglement learning is one approach to disentanglement, which hypothesizes that realistic data can be generated by a few explanatory factors of variation and these can be learned by unsupervised algorithms (Higgins et al., 2017; Burgess et al., 2018; Chen et al., 2018; Jeong & Song, 2019; Shao et al., 2020; Kumar et al., 2018). However, by design, these approaches do not incorporate supervised signals into the disentanglement process; yet such approaches typically require ground-truth factors of variation for evaluation (Higgins et al., 2017; Chen et al., 2018) making them infeasible for complex datasets where factors of variation are unknown or entangled.

Other approaches have incorporated supervised signals when learning representations, using control of input transformations (Kulkarni et al., 2015; Song et al., 2023), a cross-covariance penalty

to encourage linear independence (Cheung et al., 2015) or explicit priors and computational graph structures (e.g., Kingma et al., 2014; Sohn et al., 2015; Siddharth et al., 2017). These approaches are inflexible because they require either knowledge of input transformations or a specialized computational graph structure which is often infeasible due to feature complexity or computational constraints. Further, these approaches do not leverage existing representations and therefore require full retraining for newly proposed conditioning variables. Our goal is to re-purpose existing representations in a flexible way using flow matching because this allows efficient training of models for many different proposed conditioning variables and ultimately, enables the iterative process of discovery shown in Figure 1.

Given the limitations we have outlined above, we seek to re-purpose existing representations from pre-trained VAEs using guided latent flow matching. In particular, we use a classifier-free approach during training of the latent flow model. In the next section, we provide a brief introduction to each of these three components of our approach in WWDC.

## 2.2 METHODS

**Variational autoencoders (VAE)**: VAEs learn latent variables $z$ from data $x$ by maximizing the following amended variational lower bound (Kingma & Welling, 2022; Rezende et al., 2014; Higgins et al., 2017)

$$\mathcal{L}_{\theta,\phi,\beta} = \mathbb{E}_{q_\phi(z|x)}\left[\log p_\theta(x|z)\right] - \beta D_{\mathrm{KL}}(q_\phi(z|x) \,||\, p(z)) \tag{1}$$

$q_\phi(z|x)$ is a neural network often referred to as the encoder which is used to approximate the posterior of the latent variables $z$ given data $x$. $p_\theta(x|z)$ is also parameterized by a neural network, known as the decoder, which reconstructs the data from the latent variables. $D_{\mathrm{KL}}(q_\phi(z|x)||p(z))$ is the Kullback-Leibler (KL) divergence (Kullback & Leibler, 1951) between the encoder distribution and a prior on the latent variables, often chosen to be an isotropic unit-variance Gaussian. $\beta$ is a hyperparameter that weights the divergence.

**Flow matching**: Flow matching has emerged as a simple yet efficient framework for generative modeling by interpolating couplings between an arbitrary source distribution and target distribution; it boasts recent achievements in image generation (Esser et al., 2024; Dao et al., 2023), video generation (Davtyan et al., 2023; Polyak et al., 2025) and speech (Le et al., 2023).

A flow is a deterministic, time-continuous, bijective transformation of a $d$-dimensional Euclidean space $\mathbb{R}^d$, with extensions to other state spaces (Campbell et al., 2022; Gat et al., 2024; Chen & Lipman, 2024). A flow $\psi$ can be defined in terms of a velocity field, often referred to as a vector field, using the following ordinary differential equation (ODE) (Lipman et al., 2024):

$$\frac{d}{dt}\psi_t = u_t(\psi_t(x)), \ \psi_0(x) = x \tag{2}$$

Flow matching is based on learning the underlying velocity field $u_t$. The flow is defined by a process called simulation, i.e. solving the ODE defined in Equation 2 (Lipman et al., 2024). A probability path $p_t$ is required to interpolate between the source, $p_0$, and target distributions, $q$. A common choice for the probability path is the conditional optimal-transport (OT) or linear path with a unit Gaussian source distribution $p_0 = \mathcal{N}(0, I)$.

$$p_{t|1}(x|x_1) = \mathcal{N}(x|tx_1, (1-t)^2 I)$$

A neural network is trained to approximate the vector field that defines the flow. For Gaussian optimal-transport, the loss function for the velocity field model, $u_t^\theta$, parameterized by the model weights $\theta$, is

$$\mathcal{L}_{\mathrm{CFM}} = \mathbb{E}_{t,X_0,X_1}||u_t^\theta(X_t) - (X_1 - X_0)||^2, \tag{3}$$

where $t \sim \mathcal{U}[0,1]$, $X_0 \sim \mathcal{N}(0,1)$ and $X_1 \sim q$. With a trained velocity model, it is then possible to sample from the data distribution by solving the ODE using numerical methods (Iserles, 2008). As

the flow is bijective, it is also possible to recover the 'seed' from the base distribution that generates the target by solving the ODE in reverse from the target distribution.

**Classifier-free guidance (CFG)**: Introduced in Ho & Salimans (2021) for diffusion models, classifier-free guidance provides an efficient way of simultaneously approximating conditional and unconditional distributions (Zheng et al., 2023). This is achieved by replacing guiding information ($y$) used to produce the target with a null vector $\varnothing$ with some probability, $p_{cfg}$. This is a hyperparameter and it is recommended to be set in the range $0.1 \leq p_{cfg} \leq 0.2$ (Ho & Salimans, 2021) in order to adequately approximate the unconditional distribution, without affecting the efficacy of the conditional distribution approximation. At inference time, it is then possible to use a weighted combination of the guided and unguided velocities when simulating the ODE (Dieleman, 2022; Holderrieth & Erives, 2025).

$$u_t^{\text{CFG}} = (1 - \omega) \, u_t(x_1|x_t) + \omega \, u_t(x_1|x_t, y) \tag{4}$$

where $\omega$ is a hyperparameter that controls the guidance weight.

## 3    APPROACH

WWDC is an application of flow matching with guidance on existing manifolds of representations. We do not use the latent flow to sample latents in a traditional generative setting. Instead, we embed a sample and flow from it in reverse to the base distribution. Since the flow matching model approximates optimal transport (OT) trajectories (Lipman et al., 2024), the resulting base distribution contains structure from the original manifold, mapped to a Gaussian distribution. For example, Figure 2a shows that the class structure present in the target distribution at $t = 1$ is preserved in the base distribution at $t = 0$ presented in the middle and right panels.

We train our flow model with a finite neural network and CFG. This constrained network is optimized to map all conditional and unconditional paths from the base distribution to the target distribution. When reversed, the flow matches features that are consistent across the data distribution to similar portions of the base distribution. Guidance will alter this dynamic for the features used within the conditional guidance. For example, in Figure 2b the class structure seen in the target distribution at $t = 1$ is entirely inaccessible in the base distribution at $t = 0$. In cases such as this, where a guiding feature is fully separable, a perfect guided flow would remove the conditional feature from the base distribution.

Furthermore, we posit that more of the initial manifold's structure is maintained if that original structure matches the base distribution chosen for the flows. In an extreme example, a perfectly unit Gaussian latent distribution would already fulfill the fitting objective of a flow matching OT model with a unit Gaussian base distribution. For this work, we largely investigate VAEs, one of the most widely used neural representation learning models. As in Equation 1, the latent space of VAEs are constrained by a KL-loss which (normally) imposes some Gaussianity on the space. We chose our flow's base distributions to be Gaussian. We highlight that, due to the OT condition, the conditional flow minimally distorts structures originally mapped in the VAE's latent manifold.

In summary, our selected base and target distributions share properties because the resulting Gaussian base distribution maintains the global structure of the original data manifold. Additionally, due to multiple factors including model limitations and inherent errors in the ODE solution, the flow matching model is imperfect. The resulting distribution is therefore not a perfect unit Gaussian and some additional local structure may remain.

**Training procedure**: See Appendix A for details on each experiment and training run. For our flow matching approach, we use the Gaussian conditional optimal transport probability path to interpolate between the base and target distributions. With probability $p_{cfg}$, we drop the conditioning information $y$ used in the calculation of the vector field and replace it with a null embedding, $\varnothing$ (Ho & Salimans, 2021). This allows us to train and subsequently sample using both guided and unguided flows at inference time.

**Reverse-flow representations**: The reverse flow is produced by running the ODE solver backwards from the VAE sample at $t = 1$ until $t = 0$. We adopt the midpoint method (Süli & Mayers, 2003) for simulation at inference in this work. This produces a continuum of representations from the original VAE sample to the base distribution across time steps in the ODE solution.

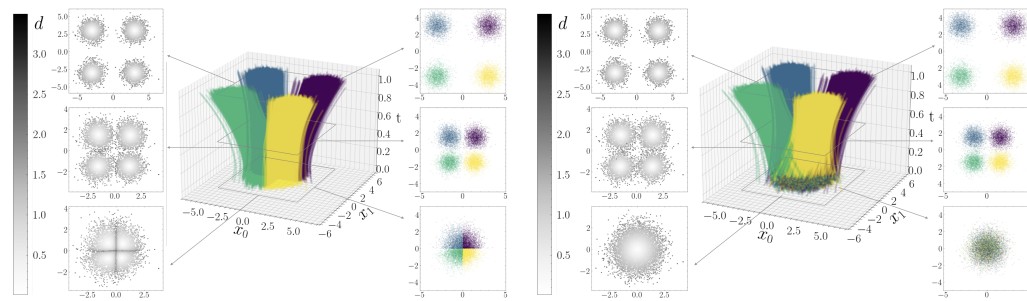

(a) Unguided flow: Class retrieval is trivial, but distance retrieval is not at $t = 0$.

(b) Class Conditional flow: Distance retrieval is trivial, but class retrieval is not (at $t = 0$).

Figure 2: Flows of the 2D Gaussian experiment detailed in Section 4.1. Each figure contains three main features: the central panel shows the flow across time steps for the 2D Gaussian class input data at $t = 1$. On either side, three 2D slices at different time steps of the flow are shown to highlight internal structures. The data points in the left slices are colored by their distance to their Gaussian's center as measured at $t = 1$. The data in the center and in the right panels are colored by their respective class index belonging to each of the four Gaussians.

## 4 RESULTS

To verify our expectations of WWDC, we explore three different datasets in increasing order of complexity and decreasing order of experimental control.

### 4.1 2D GAUSSIANS

We seed four synthetic isotropic Gaussians consisting of samples $X \in \mathbb{R}^2$ for our first experiment. For this, we train a simple flow matching model to generate the four Gaussians as the target distribution. The velocity vector field is modeled using a simple multi-layer perceptron (MLP) which takes the position $X$, a time step, and the class information as input. We use CFG with a null vector of $\varnothing = -1$.

The resulting flows are presented in Figure 2. Figure 2a shows the unguided flow, which results in a clear delineation of classes in the base distribution at $t = 0$ (see the right panels). We see that when guided on the class label, the class conditioning shows no discernible structure in the base distribution at $t = 0$ of Figure 2b.

We assign to each point a second feature: the Euclidean distance to the center of its respective Gaussian at $t = 1$. In the guided case, Figure 2b, we see that there exists a very simple map to the distance metric $d$ which is naturally apparent in the base distribution. In contrast, the structure of the classes in the unconditional case, Figure 2a, presents a much more complex distance structure.

To quantitatively consider these features, we evaluate flows at a variety of guidance weights. The results are presented in Figure 3. For the classification problem, we calculate the mutual information with respect to the class of the samples at a given time point. The results are presented in Figure 3a, which show that with a guidance weight of $\omega = 1$ (i.e. fully conditional, as presented in Figure 2), there is a turnover at $t = 0.5$ where there is almost full mutual information beyond that time, and progressively less before that time. At the base distribution, $t = 0$, there is no mutual information for a guidance weight of 1. Overall, we see that the effect of weighting the guidance is as expected: the weaker the guidance, the more of the class-wise mutual information is preserved.

We conduct a similar probe into distance metrics, simplified to a single dimension. We denote the distance from a point to its class's center along one axis or the other as $\delta_{x_i}$. We fit a linear model to predict these features across the times and guidance weights. The results are presented in Figure 3b. Because we are using a linear model, the $R^2$ score is effectively zero at $t = 1$: a given point's distance cannot be predicted with a linear model, as it is an inherently non-linear setup. However, after guiding to $t = 0$ with a weight of 1, equivalent to Figure 2b, we see increasingly better variance explainability by the simple linear model over these features until, at $t = 0$, the full dimension-wise

distance can be mapped linearly. Here the unguided, $\omega = 0$, case results in an $R^2 \approx 0.3$. This is expected as the linear explainability has certainly increased in comparison to a non-linear problem, but many samples are not correctly mapped as the resulting space is warped by the importance of class-conditional information, which has not been removed.

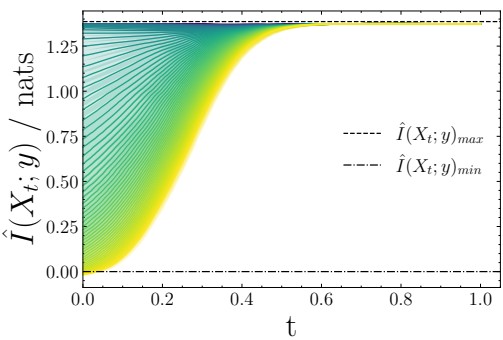 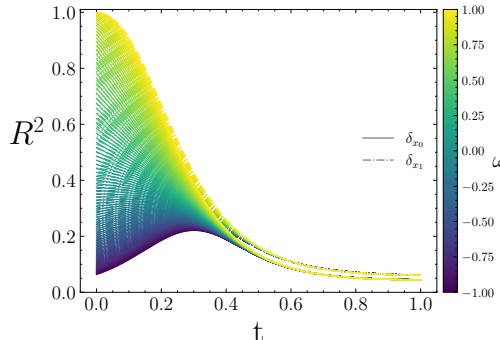

(a) Evolution of the mutual information between a distribution and the classes.

(b) Linear regressions' $R^2$ scores to dimension-wise distances to the center of the original in $x_0$ and $x_1$.

Figure 3: 2D Gaussian evaluations across guidance factors $\omega$ and time $t$. We see that at weight 1, the secondary feature is surfaced and is fully recoverable in the base distribution and conversely the class mutual information is fully gone.

## 4.2 cMNIST

As a more complex experiment, we consider colored MNIST (Deng, 2012) (cMNIST). We generate a random RGB value $(r, g, b) \in [0.05, 0.95]^3$ and multiply a three-channel version of the MNIST data by this value. We withhold $b$ from the guidance so that we can use it as a secondary feature of interest and still examine the properties of $r$ and $g$.

We first train a $\beta$-VAE on cMNIST with minimal weighting on the KL penalty weight, $\beta = 1 \times 10^{-4}$, and a latent of size 64 for improved generation quality. We then train a flow matching model on the latent space of the VAE using a simple MLP to parameterize the velocity vector field. This model is conditioned on the digit class and the maximum red and green values of the colored digit, while blue is withheld from the conditioning. Label dropout is used to estimate the unguided velocity field.

We use t-SNE (van der Maaten & Hinton, 2008) (see Appendix A.2 for more details and figures) to visualize 2D projections of the latent spaces. These are presented in Figure 4a, where we show the class of each point, and the resulting projection shows clear structure. We guide based on red, green, and digit class. The resulting $t = 0$ space is projected with the same t-SNE hyperparameters and is shown in Figure 4c. We see the class structure almost entirely disappear. We note we don't expect it to disappear entirely, as there are confounding features which are informative over class, for example: consider how expressive 'straightness' could be for predicting ones.

In tandem, we plot the same structures but colored by the strength of the blue color in the samples, a feature we didn't condition on. Here we see some, but clearly no obvious structure in the original VAE space in Figure 4b. In comparison, the guided case, presented in Figure 4d, presents a gradient across the space. It is unlikely that an annotator would find structure in the VAE space, but it is plausible that an annotator could have found the blue feature within the base conditional space.

To evaluate how guidance suppresses the digit, red, and green features, and how this enables recovery of blue, we train simple linear regression models (Pedregosa et al., 2011) on the representations. We vary the number of training examples used in these models and show the results in Figure 5. Figure 5a shows that the digit classification accuracy drops significantly with guidance in comparison to unguided and VAE representations. Similarly, in the case of a linear model regressing to the $r$ and $g$ values, the ability to recover the guiding information is repressed, and especially with fewer training samples. We see that the guidance signal does not adversely affect the sample size required to recover blue, i.e. the feature outside of the guidance, highlighting how even in this space, a few labeled samples can be quite useful.

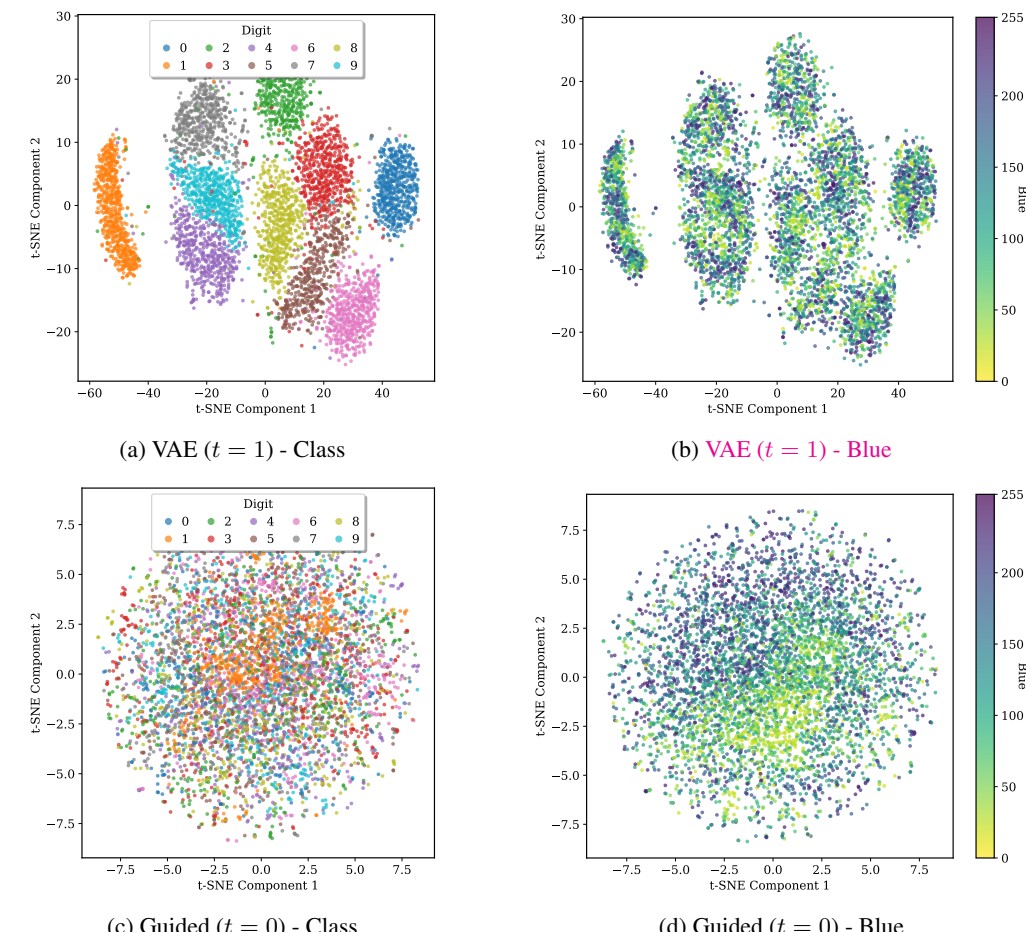

(a) VAE ($t = 1$) - Class

(b) VAE ($t = 1$) - Blue

(c) Guided ($t = 0$) - Class

(d) Guided ($t = 0$) - Blue

Figure 4: t-SNE projections of cMNIST embeddings. The VAE space in a is dominated by class structure making the task of identifying subtle features, such as blue in b, difficult. The reverse-guided flow in c removes the most visible class structure while preserving the underlying color, as shown in d, which surfaces more clearly in the structure of the new space than in b.

We note that the accuracy and $R^2$ scores not dropping by a larger degree than they have, is likely due to two factors. Firstly, as previously noted, the digit class information is inherently bound to other features that are not unique to that digit, and therefore cannot be removed with a simple class annotation (e.g. straightness or number of loops). Secondly, as we wanted high quality samples, our latent space is not as restricted as is possible for MNIST reconstruction. This was an intentional decision to enable feature interpretability and is also likely why having more training samples improves recovery of guided properties given the scale of the latent space. See training details in Appendix A.2.

Our interest in high quality reconstructions is because, as a tool, WWDC also natively enables sample generation. We first move backwards in the flow using the digit conditioning to $t = 0$, and then initialize the flow forwards using different guidance. Figure 6 shows results of this for cMNIST samples and class changes. This enables relatively cheap inspection of synthetic data. We envision this as a supporting function for the WWDC annotation loop, outlined in Fig 1, as one can conceivably inspect the same sample across different guidance to spot semantic similarities. For example, in the cMNIST case of Figure 6 we note stroke widths (e.g. 0 vs. 3), digit position (e.g. 4 vs. 7), and, of course, color are all preserved across guided samples.

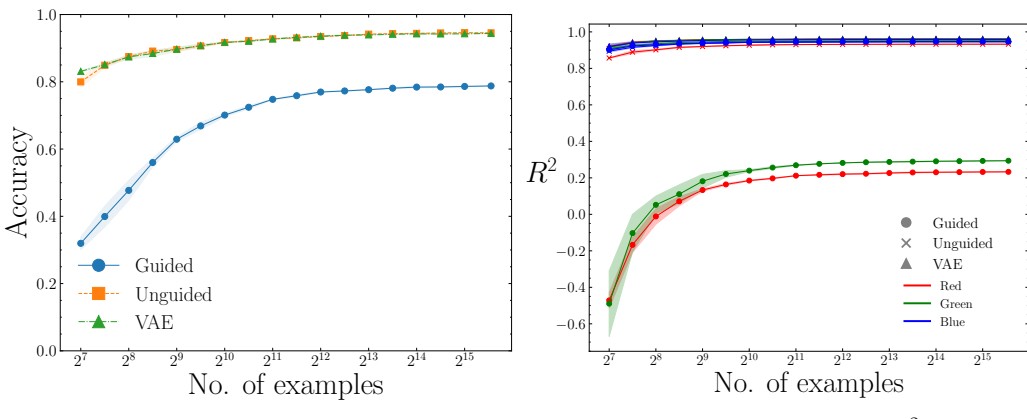

(a) Linear probe classification accuracy.

(b) Linear probe color regression $R^2$-score.

Figure 5: Linear probe evaluations for classification and regression tasks on the colored MNIST task. Note that the blue value is withheld and is consistently recovered throughout both flows.

Orig. VAE    Guided

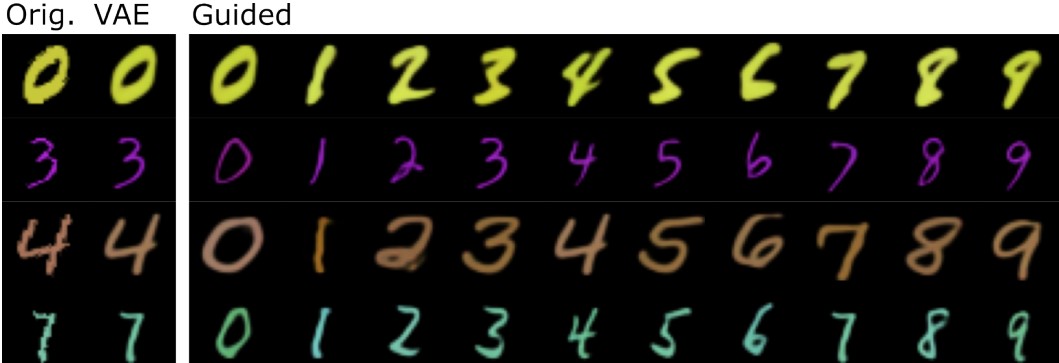

Figure 6: Style transfer in colored MNIST: The guided $t = 0$ embeddings are used to initialize a flow model but the guidance is switched, conditioning on another digit, to produce stylistically consistent digits in the VAE space.

### 4.3 Galaxy10

We now explore how techniques from previous sections are applicable in a more realistic setting. We consider scientific data, and specifically the Galaxy 10 DECaLS dataset (Leung & Bovy, 2019). Galaxies contained in the data are separated into ten broad but not necessarily distinct morphological classes. These include: disturbed, merging, round smooth, in-between round smooth, cigar round smooth, barred spiral, unbarred tight spiral, unbarred loose spiral, edge-on without bulge and edge-on with bulge. We train a VAE with a minimal KL-weighting of $\beta = 1 \times 10^{-6}$ and with latents $z \in \mathbb{R}^{4 \times 32 \times 32}$ to enable high quality reconstructions.

To demonstrate how class and concepts are disentangled during the latent flow, we first select different classes from the data and project to the base distribution using $\omega = 1$. We then flow from the base towards $t = 1$ using the 'round' class as guiding information with $\omega = 3.5$ to increase the guidance signal, following (Ho & Salimans, 2021). We choose 'round' as it is the least semantically complex structure and so enables us to view residuals of what the process has changed about the image without introducing other features from a different galaxy class. Results are presented in Figure 7. We note that the background features remain unchanged in the forward-guided image, confirming that the model has identified the galaxy of interest from extremely simple class labels and the guidance leaves these features unchanged. In sample E, we also note the imaging artifact. Here the lower half of the galaxy is yellow, which is not a physical feature but rather an imaging artifact. This feature is also preserved in the forward generated galaxy despite the structure of the galaxy being changed. Through the residuals, we see the isolation of the guided features in that image.

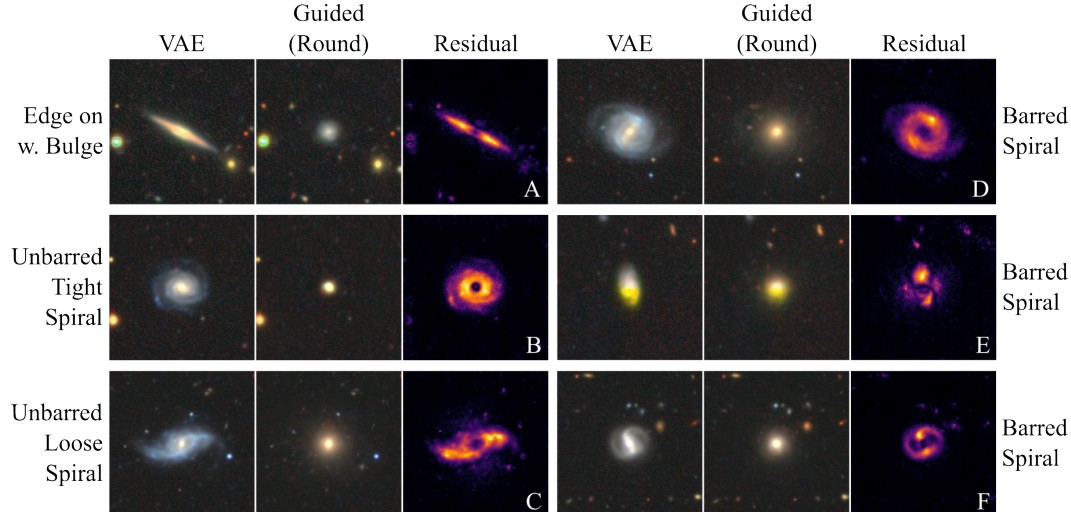

Figure 7: Samples of visual feature isolation for Galaxy10. Galaxies, their guidance-generated 'round' versions, the residuals between these images are shown. Features associated with the original galaxies are clearly separated from the remaining features of the image.

This is a native property of WWDC and could be used to understand what exactly has been captured (or not captured) by a presumptive class or measurement, adding to the scientific discovery loop envisioned in Figure 1. Importantly, separating galaxy features like this will directly enable analyses in astrophysics through modern surveys (e.g. LSST Ivezić et al., 2019) where high-dimensional complex features are abundant and whose exploration is limited largely by cost.

## 5 CONCLUSION

This work introduced What We Don't C (WWDC) as an approach to manifold disentanglement for structured discovery. Disentanglement being a distinctly difficult task, we relax the constraints on the problem to disentangling a known signal from a structured manifold with the purpose of enabling access to other features within the manifold.

We demonstrated with a toy Gaussian problem how flow models can produce *meaningful representations* of the data that surface secondary features of interest while suppressing known information. The colored MNIST experiment shows this in a more complex setting where we selectively retrieve color from latents *only* if we do not condition on them. We highlight that the latents must contain information on the style of a given digit by guided generation. Finally, we apply the method to astrophysical data, where we show that our model can disentangle important class features in galaxy images.

Critically, this approach enables reuse of existing VAEs and we expect adoption of this approach to enable structured discovery and annotation through representation learning. We identify a clear immediate use case for WWDC in assisting researchers to explore what information they haven't yet captured, either because they didn't think of it, or could not access easily.

## ACKNOWLEDGEMENT OF LLM USAGE

Large language models, including Gemini and ChatGPT, were used during the research of this paper for source finding and literature exploration. Additionally, AI assisted development tools including Cursor were used in the development of the code used in this manuscript.

## REPRODUCIBILITY STATEMENT

We include details on the experiments included in this work in Appendix A. We describe the data, models and training methods for the 2D Gaussian in Appendix A.1, color MNIST in Appendix A.2 and Galaxy 10 in Appendix A.3. The full code will be made public with a permissive license upon acceptance.

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

## A  EXPERIMENT DETAILS

### A.1  2D GAUSSIANS

The four Gaussians were each generated with covariances $\Sigma = 0.5I$ and respective means from the set $\{(\pm 3, \pm 3)^\intercal, (\pm 3, \mp 3)^\intercal\} \in \mathbb{R}$. The model that is fit to the flow is an MLP consisting of four linear layers with ELU activations in-between, resulting in 8.5k trainable parameters. We used a label dropout probability of $p_{cfg} = 0.2$.

### A.2  CMNIST

We augment the MNIST dataset Deng (2012) with color information to provide a highly controllable experiment in order to verify the intuitions we have built from the 2D Gaussian case in section 4.1. The color information is determined such that

$$c_i \sim \mathcal{U}(0.05, 0.95) \tag{5}$$

for $i = 1, 2, 3$ (i.e. the R, G and B channels of the image) and $\mathcal{U}$ denotes the uniform distribution. We also use the following data augmentations:

- Rotations: $\theta \sim \mathcal{N}(0, 10)$
- Scale: $s \sim \mathcal{N}(1, 0.1)$

The $\beta$-VAE consists of CNN encoder and decoder. The encoder and decoders are inspired by the VGG architecture (convolutions followed by batch normalization layers, ReLU, and max pooling to step down in scale). The decoder uses up-sampling instead of max pooling. There are also linear projections with ReLU activations from the CNN encoder to the latent space and a linear projection from the latent space to the CNN decoder. This model contains 23.4 M parameters.

The model is trained using $\beta = 1 \times 10^{-4}$. The flow model consists of a simple MLP of four hidden layers with $GeLU$ activations taking the latent vector and time as inputs. This model consists of 171 k parameters. The intermediate linear layers are modulated with a projection of the class information. For this, the digit is embedded and concatenated with the maximum red and green value. We then project the concatenated information using a linear layer to output a scale and shift terms for the intermediate states of the network after linear layers in a method similar to Perez et al. (2018). Guidance dropout is used with a probability of $p_{cfg} = 0.1$.

The t-SNE plots were generated with the following parameters Pedregosa et al. (2011): we used two components, perplexity was set to 100 and maximum iterations was 5000. The random state was 42 with the 'auto' learning rate setting and it was initialized using the 'pca' option.

The unguided flows produced the following projections in Figure **??**.

### A.3  GALAXY10 DECALS

The Galaxy10 DECaLS Dataset contains $17\,736$, $256 \times 256$ sized colored galaxy images in the g, r and z band which have been scaled for clarity to RGB PNGs. These are separated in ten broad classes including: disturbed, merging, round smooth, in-between round smooth, cigar round smooth, barred spiral, unbarred tight spiral, unbarred loose spiral, edge-on without bulge and edge-on with

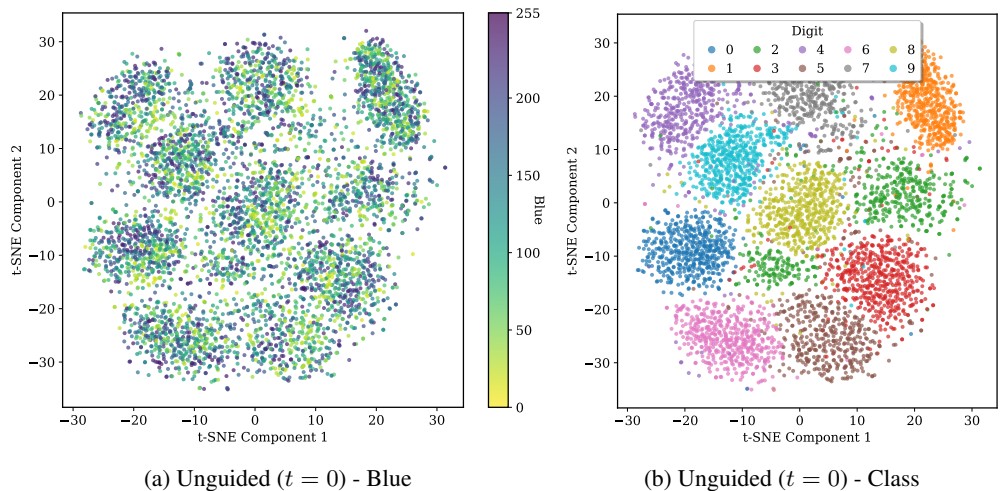

(a) Unguided ($t = 0$) - Blue

(b) Unguided ($t = 0$) - Class

Figure 8: t-SNE projections of cMNIST embeddings for the unguided distributions. The reverse-guided flow in 8b retains the class structure of the VAE. Figure 8a shows how the structure of the blue color is difficult to observe in the unguided space.

Orig.    VAE    Guided

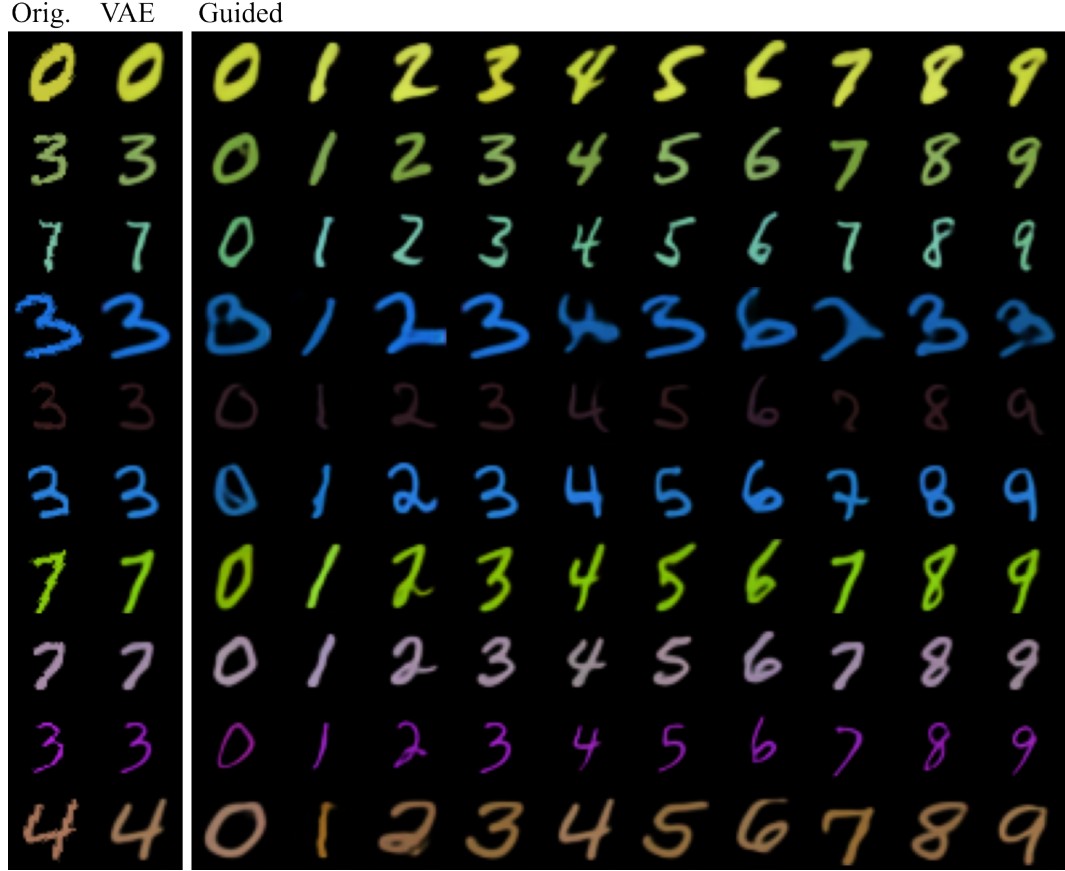

Figure 9: More examples of style transfer using the guided base representation.

bulge. The labels were originally provided by volunteers from the Galaxy Zoo project (Lintott et al., 2011) and the collection was compiled by Leung & Bovy (2019).

Orig. VAE Guided

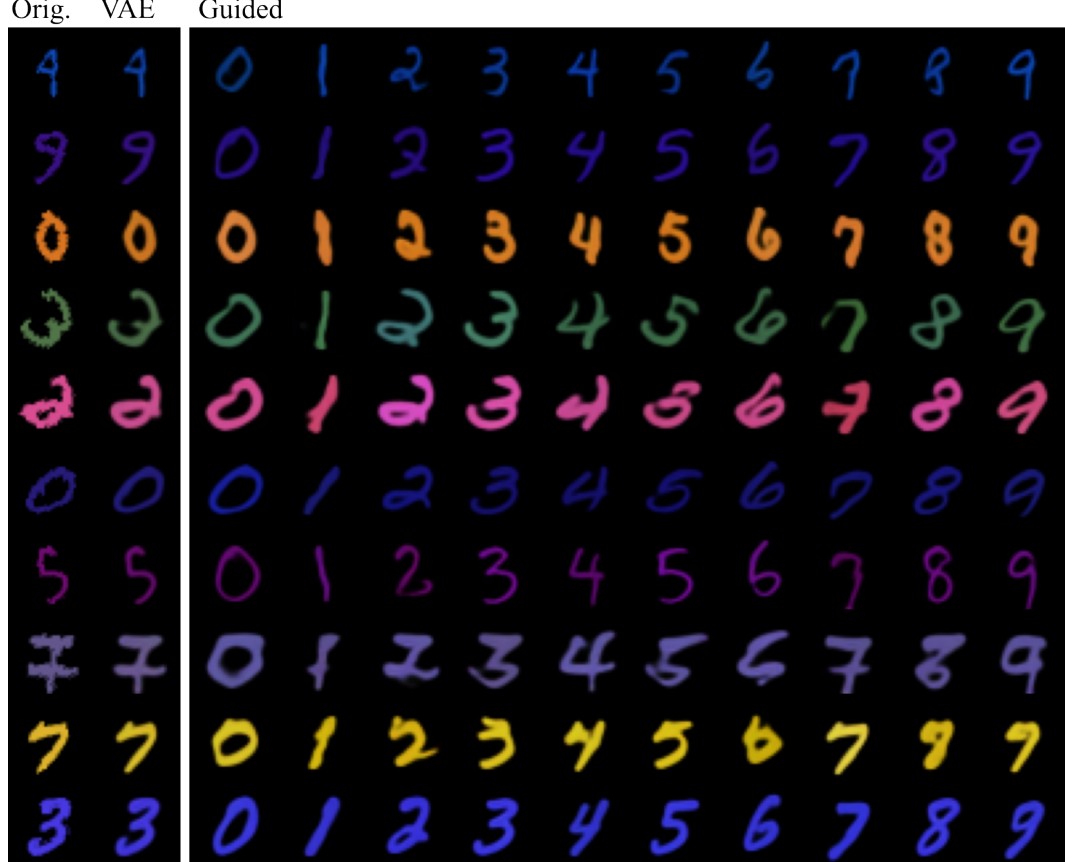

Figure 10: Examples of style transfer using the unguided base representation. Although these examples are close to the original VAE digit, they do not produce the level of correspondence seen in the guided base samples used in figure **??**.

A $\beta$-VAE with $\beta = 1 \times 10^{-6}$ was trained on the images to produce a $4 \times 32 \times 32$ latent representation. We used the diffusers (von Platen et al., 2022) implementation of a variational autoencoder with four down sampling blocks using the "DownEncoderBlock2D" in the encoder, each outputting 32, 64, 128 and 256 channels respectively. The decoder follows a symmetric structure using "UpDecoderBlock2D". Each block in the VAE has four layers. This model accounts for 20.3M parameters.

We use a class conditional U-Net from the TorchCFM package (Ronneberger et al., 2015; Tong et al., 2024a;b) to parametrize the velocity field using the galaxy10 classes as the conditioning signal. The U-Net has four layers with 64, 128, 128 and 128 channels. Each down sampling of the U-Net has a single residual block. This totals 6.1M parameters. Conditioning dropout is used with a probability of $p_{cfg} = 0.1$.

## B  RELATED WORK

Useful low-dimensional representations of data can be recovered by various techniques. These include linear methods (e.g. PCA; Pearson, 1901), non-linear methods (e.g. t-SNE; van der Maaten & Hinton, 2008), and deep learning methods (e.g. variational autoencoders; Hinton & Salakhutdinov, 2006; Kingma & Welling, 2022; Rezende et al., 2014). In addition, self-supervised learning (SSL) methods have been shown to produce useful representations for downstream tasks, including approaches such as SimCLR (Chen et al., 2020), BYOL (Grill et al., 2020), JEPA (Assran et al., 2023), and LeJEPA (Balestriero & LeCun, 2025). Various works have investigated best practices in extracting meaningful or semantic representations from models trained using different methods across domains. This includes student teacher based systems (Grill et al., 2020; Oquab et al., 2023),

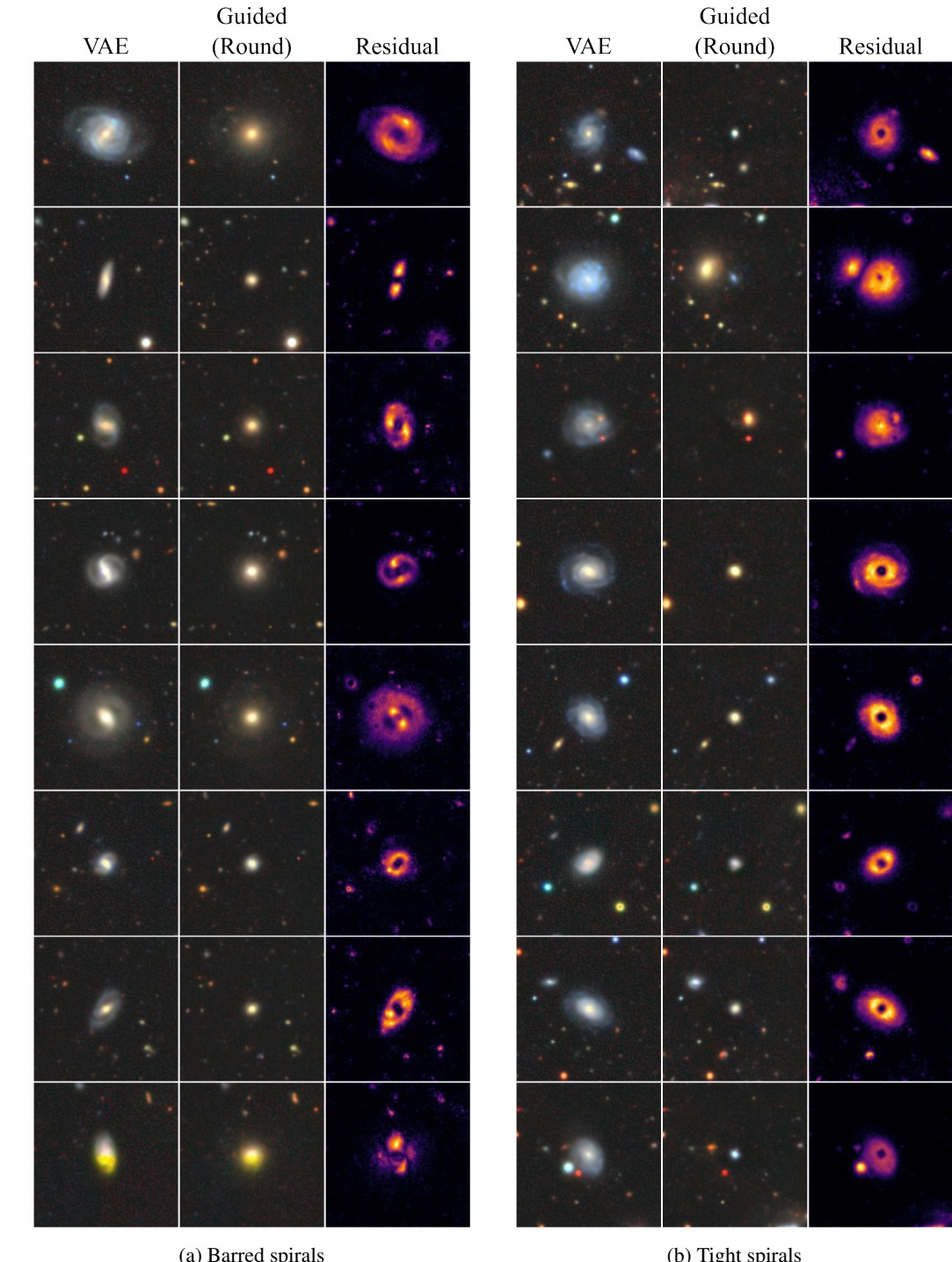

|          | Guided  |          |          | Guided  |          |
| VAE      | (Round) | Residual | VAE      | (Round) | Residual |

(a) Barred spirals                                                            (b) Tight spirals

Figure 11: Barred vs. unbarred tight spirals.

masked modeling (He et al., 2021), auto regression (Tao et al., 2024), and diffusion (Tang et al., 2023; Fuest et al., 2024; Luo et al., 2023).

Constraining the structure of a latent space has become common practice in representation learning. This is particularly true in unsupervised disentanglement learning which hypothesizes that realistic data can be generated by a few explanatory factors of variation. For example, many approaches

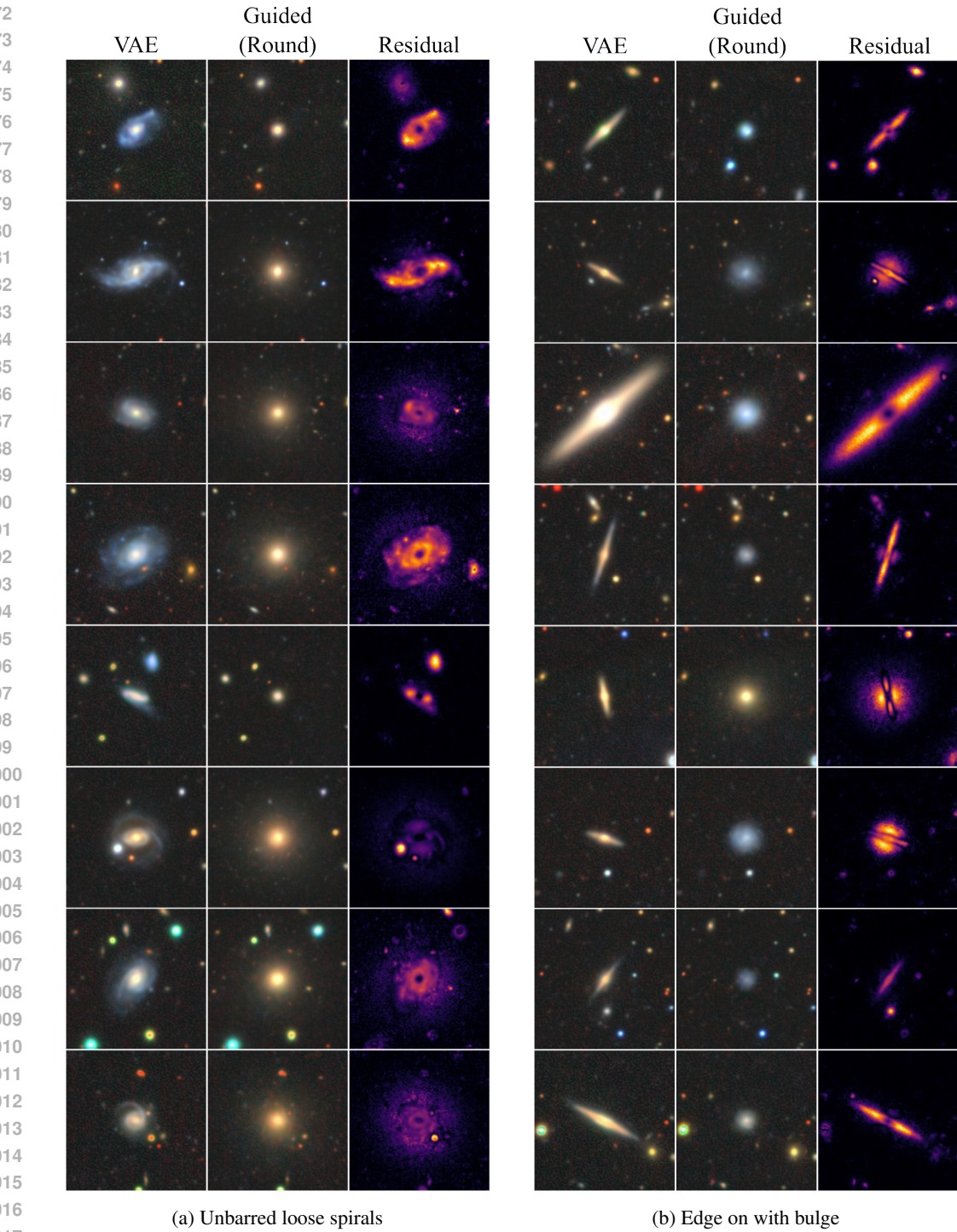

|  | Guided | | | Guided | |
| VAE | (Round) | Residual | VAE | (Round) | Residual |

(a) Unbarred loose spirals          (b) Edge on with bulge

Figure 12: Additional feature isolation examples.

implicitly or explicitly penalize the total correlation between latent variables in order to encourage disentanglement (Higgins et al., 2017; Burgess et al., 2018; Chen et al., 2018; Jeong & Song, 2019; Shao et al., 2020; Kumar et al., 2018). However, Locatello et al. (2019) have shown theoretically that unsupervised learning of disentangled representations is impossible without inductive biases on the models or data. They also show empirically across many different unsupervised methods, that although the aggregated posterior of latent distributions are uncorrelated, the means of these

distributions used for downstream tasks are often correlated. Additionally, unsupervised approaches typically require ground-truth factors of variation for validation (Higgins et al., 2017; Chen et al., 2018).

Supervised control over input transformations has been used to create useful representations (Kulkarni et al., 2015; Song et al., 2023). However, such approaches generally require well-defined transformations of the input data and may not be available and may not span all data features. Other works have incorporated supervised signals when learning representations. Cheung et al. (2015) used a cross-correlation penalty between available information and latent variables learned by an autoencoder to encourage *linear* disentanglement between subspaces. This approach introduces additional terms to the reconstruction loss including a supervised and a cross-covariance term between the latent and the supervised component and therefore requires additional hyperparameters to ensure each objective is balanced and achieved during training. Finally, if new conditioning information is discovered or introduced, the full model requires re-training with an amended structure making this expensive for processes which seek to control and refine representations iteratively.

Other approaches have extended variational autoencoders to include additional conditional information (e.g., Kingma et al., 2014; Sohn et al., 2015; Siddharth et al., 2017). Such methods require the definition of priors on the conditioning information and are incorporated into the structure of the model. Additionally, the variational objective can be augmented to include (semi)-supervised losses. Training these models requires balancing the supervised loss terms with the original variational objective, requiring hyperparameter searches. This becomes even more difficult if the number of conditioning variables grow. As discussed in previous methods, if new conditioning information is discovered, the full model requires re-training with a new structure making this an inflexible approach if new or more refined conditioning information becomes available.

## C LIMITATIONS

This work is in early development and so it should be noted that there are a number of limitations. These include:

- Solutions to ODEs used for the flow model training and inference introduce an inherent source of error. At this stage no quantification has been undertaken on how this may impact the representations during the chain, especially with the loss of information in the conditioning variables. More work is needed to understand how the capacity of the velocity field network, the simulation at inference and optimization procedures can aid or hinder representations.

- Computational resources were limited in the development of this paper, and therefore a full investigation into the hyper-parameters associated with the flow model training has not been undertaken. This is especially true of the latent size of the VAE and the dropout frequency used in training and how it may impact the quality of the unguided distributions.

- It is unclear which conditioning mechanisms are the most appropriate and efficient in approximating the guided velocity field and further work is needed to find the most effective guidance mechanisms.

- Currently we only consider a Euclidean state space $\mathbb{R}^d$. More work is needed to understand whether this approach works on other state spaces such as discrete tokens and quantized VAEs.

