# OpenReview forum: "What We Don't C: Manifold Disentanglement for Structured Discovery"
_ICLR.cc/2026/Conference — Submitted to ICLR 2026_

### Official Review · Reviewer_GFk4 · 2025-11-01

**Soundness:** 3
**Presentation:** 3
**Contribution:** 2
**Rating:** 6
**Confidence:** 2

**Summary:**

This paper introduces What We Don’t C (WWDC), an approach aimed at uncovering residual factors of variation in latent spaces by intentionally removing information associated with known conditioning signals (e.g., class labels). The method starts from a pretrained VAE, then uses latent flow matching with classifier-free guidance to subtract the influence of a chosen conditioning signal. By reversing that guided flow, the model produces a latent representation where that signal is minimized, making it easier to see what other structure was hidden underneath.

**Strengths:**

The paper has a clear and compelling motivation: instead of continuing to reinforce information we already understand in a dataset, it focuses on uncovering what remains after known factors are removed. This conceptual reframing is refreshing and feels genuinely useful, especially for exploratory scientific analysis. The authors present the idea in an intuitive way, and the progression of experiments (from synthetic data to real astrophysics imagery) helps build trust in the approach. The qualitative results are effective in showing how the method reveals subtle structure in the data that wasn’t obvious before. Additionally, the fact that WWDC operates on top of pretrained models makes it practical and easy to adopt in real workflows, instead of requiring heavy retraining or specialized architectures.

**Weaknesses:**

The main limitation is that the evaluation remains largely qualitative, making it difficult to assess how well the method performs relative to established baselines in representation learning or disentanglement research. The paper would benefit from more systematic quantitative comparisons or metrics to support its claims. Some of the theoretical explanations around how information is preserved or removed during the latent flow process are also hard to follow and could use clearer intuition rather than relying primarily on equations. Finally, while the galaxy experiment is visually compelling, the paper does not fully explore the robustness or generality of the method across other complex real-world domains, which leaves questions about how broadly applicable the approach truly is.

**Questions:**

1. How sensitive is WWDC to the choice of pretrained VAE or latent dimensionality? The method is demonstrated on a particular architecture setup, but it's unclear whether the residual patterns remain stable if the base model or latent size changes. Some ablation or robustness analysis would help clarify whether the effect is consistent or dependent on specific model configurations.

2. Can you quantitative evaluation for the “emergent residual factors” revealed by WWDC? While the visualizations are compelling, it would strengthen the work to include metrics showing how well the extracted residual structure correlates with meaningful latent attributes (e.g., cluster separability, predictive power on downstream labels, or mutual information comparisons).

---

> ### Author Response · Authors · 2025-11-20
> **Response to Reviewer GFk4**
>
> We thank reviewer GFk4 for their review of the manuscript. We have made substantive revisions to improve the readability of the manuscript, including addressing this review’s concerns around the clarity and interpretability of the theoretical arguments. We thank the reviewer for this. We hope this greatly improves the manuscript and that it is now much more readable by a wider audience. We respond to the reviewer’s points explicitly below.
>
> ## Weaknesses
>
> We address each sentence in the paragraph under the weaknesses section in order below:
>
> **W1**: So far we have found existing baselines used in disentangled representation learning, such as the disentanglement metric score (Higgins et al. 2017) or mutual information gap (Chen et al. 2018), as insufficient for our method. This is because these metrics typically measure ‘latent-by-latent’ disentanglement properties. Other supervised approaches, as in Siddharth et al. 2017 or Cheung et al. 2015, make extensive use of qualitative measures of disentanglement for the representations they learn through generative control.
>
> **W2**: We agree with the reviewer that systematic quantitative metrics and potential comparisons would help strengthen the manuscript. This is a work in progress and requires well-informed benchmarks; we believe this is non-trivial given that our approach to disentanglement and discovery differs significantly from existing methods.
>
> **W3**: We have updated the *Background* and *Approach* (L179–206) sections of the paper to make our theoretical arguments clearer. In addition, we have added more detail to the caption of *Figure 2* and additional description to the results in *Section 4.1*, in particular L256–263, which outlines the mutual information evolution during the latent flow for various guidance weights. We hope the revised *Section 4.1* augments the clarified theory arguments developed in the revised *Approach* section, in order to provide a more intuitive explanation for how information is preserved or removed during the latent flow process.
>
> **W4**: Our current work is exploring how the methodology and results from this paper can be applied in additional complex real-world domains in a robust manner. In particular, we are excited about more general applications beyond the Galaxy experiment shown in this work.
>
> ## Questions
> **Q1**: We intentionally used different scales of embedding spaces across the three datasets and their respective experiments. Due to computational limitations, we were unable to do a full exploration of this hyper-parameter as much as we would have liked. We recognise this as a limitation in the *Limitations* section in Appendix C. We note that there are natural trade-offs that are well known on this scale, and we state in the paper that we are especially interested in using VAEs as generative tools to guide data exploration for a given data annotator, as in Figure 1. Having high quality reconstructions demands a sufficiently expressive encoder-decoder architecture, and due to computational limitations we have not yet been able to fully explore these trade-offs.
>
> **Q2**: We have updated parts of our results section to make the relevant evaluations more salient. We more clearly present results on mutual information and regression metrics for the 2d Gaussian case in *Figure 3*. We also more clearly present results on the cMNIST classification accuracy and color regression in *Figure 5*.
>
> We thank the reviewer again for their constructive and positive feedback. Their feedback has helped improve the manuscript and we look forward to further discussion.
>
> Please note that we have highlighted all of our edits in the manuscript to reduce the burden on the reviewer and will be altering the color to match the text for the camera-ready version.

---

### Official Review · Reviewer_M6hy · 2025-11-01

**Soundness:** 2
**Presentation:** 2
**Contribution:** 2
**Rating:** 4
**Confidence:** 2

**Summary:**

The paper proposes a new method for disentanglement of the latent spaces of supervised (e.g., with class labels) generative models, using flow matching.

**Strengths:**

- addresses an important problem in an interesting way (including allowing further disentanglement of pretrained models)
- reasonable breadth of experiments, from simple controlled to complex real datasets, including intuitive results in figures 6 and 7
- intuitive results, especially in figures 6 and 7

**Weaknesses:**

- no reproducibility statement or opensource code, which is especially important for less theoretical contributions like this
- no (argument for the lack of) clear contextualization or comparison against existing disentanglement approaches
- hard-to-follow theory presentation in sections 2 and 3; maybe I just lack the background, but I guess I'm not the only reader who would benefit from gentler, more precise guidance through it
- unpolished writing

**Questions:**

My main questions, following from the weaknesses above, are:
1. can the authors can precisely explain and formalize the notion of disentanglement being used?
2. having done that, can the authors motivate this notion? Is it just "yet another" disentanglement notion, or is it somehow more fundamentally a better notion that others in the literature?
3. and can the authors relate this to other formalizations of disentanglement, including experimental comparisons where appropriate?

Less important, but still helpful for me to better understand the work:
1. can the authors rephrase the sentence starting at L085? I don't really understand it or even how to parse it.
2. L212: what does "ideal" mean here? Is it not ideal just in practice, or also at a more fundamental level?
3. L234: how do Markov chains come into play here? Is this related to eq. (2)?
4. L238: what does "information is not a sufficient criteria to enable access to that information" mean?
5. L276: what expected structure "is not obvious" here? Is it obvious in the right panel? I don't understand what I'm supposed to be seeing here.
6. L278: (related to my main questions about disentanglement above,) why is it desirable that "the class information has been entirely remove" here?
7. L317: what's "minimal weighting" mean here? Is this some hyperparameter selection method?
8. L347: what's the "clear pattern across the space" here? I'm not sure I see it.

Typos:
- L029: the YM reference here is formatted incorrectly (also in the bibliography), mixing up last name and fist/middle initials.
- L080: missing space "models(Fuest"
- L087: "to to"
- L138 and beyond: math formatting could be improved, e.g.,
    - $D_{\mathrm{KL}}$
    - $\mathcal{L}_{\mathrm{CFM}}$
    - $p_{\mathrm{cfg}}$
    - ${u_t}^{\mathrm{CFG}}$
- L174: "trained used"
- L207: VAEs -> VAE's
- L305: "features.Figure"
- L313: "a random RGB values"
- L238: "a sufficient criteria" ("criteria" is plural; the singular is "criterion")
- L444: "Uunbarred"
- L466: "we propose to be used to"
- L467: "surveys(e.g."
- L478: "representations of the data that emerge secondary features"
- L485: "in assisting researchers explore what information"

---

> ### Author Response · Authors · 2025-11-20
> **Response to Reviewer M6hy**
>
> We thank Reviewer M6hy for their time and their careful reading of the manuscript. We have made substantial changes to the manuscript based on this review. Our manuscript is greatly improved by their comments and questions. We address each point individually for clarity.
>
> ## Weaknesses
>
> **W1**: We have added a reproducibility statement in L486. The full code will be made public with a permissive license upon acceptance.
>
> **W2**: We have clarified why it is not appropriate for us to compare ourselves directly to other disentanglement approaches in L092–L120. Explicitly, our proposed manifold-disentanglement paradigm is fundamentally different from traditional disentanglement learning, which aims to disentangle features into individual axes (Higgins et al. 2017) or through latent functional mappings (Song et al. 2023). WWDC is intended only to condition on known manifold features, allowing other manifold features to become more accessible.
>
> **W3**: We have thoroughly revised the *Introduction*, *Background*, and *Approach* sections to present a more approachable introduction to the methodology.
>
> **W4**: We have thoroughly reviewed and edited the manuscript to improve the editorial standard.
>
> ## Questions
>
> **Q1**: We have added a clearer formulation in L092–L120 in *Section 2.1* and in our responses to the weaknesses comments above.
>
> **Q2**: The differences to traditional disentanglement approaches (clarified above) enable WWDC not to assume that the information is fully separable into unique dimensions or paths on a manifold. Rather, we assume nothing about the structure of the underlying signal. Our lack of assumptions makes this approach more flexible than augmentation-based disentanglement and more useful than $\beta$-VAEs or other total-correlation disentanglement methods, which can struggle on real data.
>
> **Q3**: Disentanglement learning has been dominated by unsupervised approaches which propose that 'data is generated by a few explanatory factors of variation' (Locatello et al. 2018). For example, $\beta$-VAEs, which propose that such factors of variation can be recovered by exploring the meaning of each latent dimension.  Additionally, few works have explored how to capture representations from pre-trained VAEs using existing knowledge about the underlying data.
>
> **L085**: We have rephrased this sentence to be easier to read with a broader audience in mind. See L034–L044.
>
> **L212**: Here, 'not ideal' refers to the fact that there are computational errors and fit errors that an OT-based flow exhibits in practice, in comparison to the analytic solution. We have clarified this in L202–206.
>
> **L234**: The flow matching model defines a Markov chain between the base and target distributions. It is related to *Equation 2* because the initial state determines the full trajectory. We link this to arguments from information theory (Cover and Thomas 2006), that information is preserved in a Markov chain, unless acted upon by a conditioning mechanism.
>
> **L238**: We want to remind readers that just because there 'is information' available does not mean that information is structured in any particular way. For an extreme example of this, see Boué 2019.
>
> **L276**: We have rephrased for clarity in the paragraph beginning L264. We were referring to the “not obvious” presentation of the  distance $d$ in *Figure 2a*, left panel at $t=0$. Although there is structure, it is not a particularly natural one, and so we referred to it (imprecisely) as “not obvious”. In comparison, in the left panel of *Figure 2b*, the $d$ colored data shows that a scaled radial measurement would return $d$ directly.
>
> **L278**: It is not necessarily desirable on its own. The thesis of our manifold disentanglement is that we can preserve global manifold structures while removing strong signals which could dominate. Not removing the classes in *Figure 2a* produces a space that is dominated by class, and $d$ is non-trivially encoded. When we remove the class signal in *Figure 2b*, $d$ is encoded in a very clear structure. This surfacing of new features is what makes it desirable to remove class information in this setting. We have also clarified this in the text in L264–273 and the caption of *Figure 2*.
>
> **L317**: ‘Minimal weighting’ refers to the $\beta$ term in the VAE pre-training. We have now stated this explicitly in L330 and L417.
>
> **L347**: The ‘clear pattern’ refers to the color map gradient in *Figure 4d*, where we see low blue values in the bottom-central portion of the plot, and higher blue values towards the edges. We have included the blue color map for the VAE embeddings in *Figure 4b* to assist the reader in seeing this contrast.
>
> Finally, we have addressed all typos. We thank the reviewer again for actively working to improve this manuscript.
>
> Please note that we have highlighted all of our edits in the manuscript to reduce the burden on the reviewer and will update the color to match the main text for the camera-ready version.

---

### Official Review · Reviewer_E5oq · 2025-11-02

**Soundness:** 2
**Presentation:** 3
**Contribution:** 2
**Rating:** 2
**Confidence:** 3

**Summary:**

In this paper, the author proposes applying flow matching to disentangle the latent space. The main method is designed for variational autoencoders (VAEs), whose latent space approximates a normal distribution, allowing the use of Gaussian conditional optimal transport probability paths for interpolation.

**Strengths:**

1. The idea of combining flow matching with variational autoencoders (VAEs) is interesting and has potential to inspire further exploration in disentangled representation learning.

2. The paper is well-structured.

**Weaknesses:**

1.  Since the method is built on top of VAEs and relies on the approximately Gaussian distribution of their latent space, its use is restricted to this specific class of generative models.

2. The paper lacks sufficient supporting evidence. In the experimental section, the author evaluates the method on synthetic 2D Gaussian data, CMNIST, and a real-world dataset. All three datasets are relatively simple, and other existing disentanglement methods are known to perform well on them—particularly on 2D Gaussian and MNIST, which are common benchmarks. More importantly, the paper does not include comparisons between the proposed approach and other established disentanglement methods, which makes it difficult to assess the effectiveness and advantages of the method.

3. If the key claim of the paper lies in achieving meaningful disentanglement, there appears to be no theoretical guarantee or clear intuition explaining why such disentanglement can be discovered using the proposed approach.

**Questions:**

1. What is the most significant difference between the proposed method and other state-of-the-art disentanglement approaches that makes it stand out? Is there any theoretical justification or empirical evidence supporting this distinction?

2. Can this method be extended or adapted to other types of generative models beyond VAEs?

---

> ### Author Response · Authors · 2025-11-20
> **Response to Reviewer E5oq**
>
> We thank Reviewer E5oq for their time and constructive review. Our writing could have been clearer in setting out our work; we have made a substantial effort to improve the clarity of the manuscript. In doing so, we have restructured both the *Introduction* and *Approach* sections. We hope that this makes reading the manuscript less confusing and improves readers’ understanding of the purpose, scope, and details of the work presented in this work. We have also made a concerted effort to clarify points associated with each of the listed weaknesses in the updated text. We explicitly respond to each weakness and question below.
>
>
> ## Weaknesses
>
> **W1**: The updated *Approach* section clarifies this point. Specifically, although we use VAEs because of their widespread use and inherent Gaussian constraints, any reverse flow to any base distribution using optimal transport will retain structure from the target manifold. Gaussianity provides more stable constraints but is not required for this method.
>
> **W2**: We recognise that our description of our approach, and its purpose, was not particularly clear. Our revised *Introduction* should help address this. Explicitly, this manifold-disentanglement paradigm is fundamentally different from traditional disentanglement learning, which aims to disentangle features into individual axes (e.g. Higgins et al. 2017) or disentangle through some function mapping in a latent manifold (e.g. Song et al. 2023). WWDC is intended only to condition on information about known features from a manifold, to allow other features of the manifold to become more accessible. As we state in the revised *Introduction*: ‘approaches to representation learning often remain unchanged even when supervised labels are available’ (see L032). The purpose of WWDC itself is not disentanglement learning as is commonly found in the relevant literature. Because of these fundamental differences between the methods, we are not aware of a clear metric that can be used to compare our approach to the literature.
>
> **W3**: We recognise the unfortunate framing centred strongly on disentanglement. We have clarified the writing in the paper to be clear that “manifold disentanglement” is not the same as the tasks proposed in disentanglement learning literature. We have rewritten our *Approach* section to give a better intuition for why a structured manifold that is missing features associated with the conditional variable can be uncovered through guided latent flow matching.
>
> ## Questions
>
> **Q1**: At its core, the problem statement is very different from disentanglement learning literature. We have clarified this in the updated manuscript (see L092–L120). In practice they are so different in terms of use, scope, and utility, that we do not believe substantial comparisons are practical.
>
> **Q2**: We lay out how the underlying components of the method are present irrespective of VAEs. OT-based latent flow matching will always map structure from target manifolds into the chosen base distribution (which need not be Gaussian). We choose to focus on VAE latent spaces because of their prolific use in the field, their wide availability, and their relatively low cost compared with models such as pre-trained LeJEPA. Using a Gaussian base distribution is also a choice we make based on the engineering simplicity and the alignment with the standard VAE constraints.
>
> We thank the reviewer again for their constructive feedback. We look forward to an active discussion on our comments and revisions addressing all of the reviewers listed weaknesses and questions.
>
> Please note that we have highlighted all of our edits in the manuscript to reduce the burden on the reviewer and will be altering the color to match the text for the camera-ready version.

---

### Author Response · Authors · 2025-12-02
**Comment to the Area Chair**

During the rebuttal process, we identified that the initial reviews contained misunderstandings stemming from our original manuscript. We recognise the reviewers’ concerns, and have therefore worked to identify opportunities to improve our communication. We have made substantial changes to enhance the clarity of the manuscript. We believe that the manuscript has been greatly improved during this process; we have outlined the changes extensively in our responses to the reviewers, as well as explicitly marking and referencing these changes in the text.

The reviewers noted, in particular, that benchmarking against other methods in disentanglement learning was desirable. This could have been avoided with clearer communication in the manuscript. First, we clarified our aim and the meaning of ‘disentanglement’ as adopted in this manuscript. We also elucidated how it differs from existing unsupervised approaches that dominate the literature (i.e., Higgins et al., 2017; Chen et al., 2018). Second, we pointed to more appropriate comparisons of our work that rely heavily on qualitative measures to assess the quality and properties of resulting conditional distributions using style transfer plots (e.g., Siddharth et al., 2017; Cheung et al., 2015).  We believe that both aspects are important in addressing any previous concerns the reviewers had regarding benchmarks.

We have comprehensively addressed the reviewers’ comments regarding weaknesses during the first stage of our rebuttal, and we have answered all of their questions. In each of our rebuttal comments, we have clearly outlined our responses to every aspect of the reviews and referenced changes in the text where appropriate. We appreciate the detailed feedback the reviewers provided in improving the quality of the manuscript.

---

### Meta-Review · Area_Chair_dZMA · 2026-01-07

**Summary:**

Reviewers raised concerns primarily about the lack of empirical comparisons and quantitative evaluation. Reviewer **E5oq** noted that the paper does not include comparisons with existing disentanglement approaches, including conditional or supervised disentanglement methods. While the authors explained that their goal differs in that they aim to extract information using prior knowledge (e.g., supervised labels), the absence of direct comparisons remains a concern. Reviewer **M6hy** shared a similar concern, emphasizing that even if the objective differs, empirical comparisons are still necessary to contextualize the contribution.
Reviewer **GFk4** additionally pointed out that the evaluation is largely qualitative and lacks quantitative support. Although the authors argued that existing disentanglement metrics may not be suitable for their approach, no alternative quantitative metrics were introduced, leaving the evaluation primarily qualitative.

Overall, these concerns about missing comparisons and insufficient quantitative evaluation informed the suggested decision.

**Reviewer Concerns:**

### Addressed

1. **Presentation and reproducibility:**
   Typos were corrected, the reproducibility statement was added, and the overall writing was polished.

### Outstanding

1. **Lack of comparisons with existing methods:**
   The paper does not include comparisons with existing conditional disentanglement approaches, such as conditional nonlinear ICA.

2. **Lack of quantitative evaluation:**
   The evaluation remains largely qualitative, without sufficient quantitative metrics to support the claims.

**Reviewer Scores:**

Reviewer **E5oq** raised concerns about the lack of experimental comparisons with existing disentanglement approaches. The authors explained that their goal is to extract information using prior knowledge (e.g., supervised labels), rather than to perform standard disentanglement. However, no comparisons were provided, even with conditional or supervised disentanglement methods. As a result, this reviewer is likely to remain negative.

Reviewer **M6hy** raised a similar concern regarding the absence of comparative experiments. Although the authors clarified that their method is intended to recover information from known manifold features, empirical comparisons were still not included. Consequently, this reviewer is also likely to remain negative.

Reviewer **GFk4** expressed concern that the evaluations are largely qualitative and lack quantitative results. The authors argued that existing metrics may not be suitable for their approach; however, no alternative quantitative metrics were introduced. As a result, the evaluation remains primarily qualitative, and this reviewer may maintain or lower their score.

---

### Decision · Program_Chairs · 2026-01-26

Reject